# Structural and Functional Analysis of Disease-Linked p97 ATPase Mutant Complexes

**DOI:** 10.3390/ijms22158079

**Published:** 2021-07-28

**Authors:** Purbasha Nandi, Shan Li, Rod Carlo A. Columbres, Feng Wang, Dewight R. Williams, Yu-Ping Poh, Tsui-Fen Chou, Po-Lin Chiu

**Affiliations:** 1Biodesign Center for Applied Structural Discovery, School of Molecular Sciences, Arizona State University, Tempe, AZ 85287, USA; pnandi@asu.edu; 2Division of Biology and Biological Engineering, California Institute of Technology, Pasadena, CA 91125, USA; lflshan@caltech.edu (S.L.); rodcarlo.columbres@gmail.com (R.C.A.C.); fengwang@caltech.edu (F.W.); yu-ping.poh@asu.edu (Y.-P.P.); 3Eyring Materials Center, Arizona State University, Tempe, AZ 85287, USA; dewight.williams@asu.edu

**Keywords:** p97 ATPase, p97^R155H^ mutation, p47 cofactor, arginine finger, IBMPFD, single-particle cryo-EM

## Abstract

IBMPFD/ALS is a genetic disorder caused by a single amino acid mutation on the p97 ATPase, promoting ATPase activity and cofactor dysregulation. The disease mechanism underlying p97 ATPase malfunction remains unclear. To understand how the mutation alters the ATPase regulation, we assembled a full-length p97^R155H^ with its p47 cofactor and first visualized their structures using single-particle cryo-EM. More than one-third of the population was the dodecameric form. Nucleotide presence dissociates the dodecamer into two hexamers for its highly elevated function. The N-domains of the p97^R155H^ mutant all show up configurations in ADP- or ATP*γ*S-bound states. Our functional and structural analyses showed that the p47 binding is likely to impact the p97^R155H^ ATPase activities via changing the conformations of arginine fingers. These functional and structural analyses underline the ATPase dysregulation with the miscommunication between the functional modules of the p97^R155H^.

## 1. Introduction

Human p97/VCP (valosin-containing protein) belongs to the type II AAA+ (ATPase Associated with various cellular Activities) protein family, which has two AAA+ ATPase domains [1,2]. The orthologs are known as Cdc48 in *Saccharomyces cerevisiae* and transitional endoplasmic reticulum (TER) ATPase in archaebacteria and eukaryotes. p97 acts as a molecular hub, interacting with various cofactors to perform a wide variety of cellular functions, including autophagy, cell-cycle regulation, ubiquitin-dependent proteostasis, and reassembly of Golgi and nuclear membranes [3,4,5,6,7]. p97 is abundant in the cytosol, comprising 1% of the cytosolic proteins [8,9], and carries out ATP hydrolysis to gain energy to fuel the conformational change underlying its activities [10,11,12,13]. Abnormal functions or uncontrolled regulations of p97 can cause serious diseases. Single amino acid mutations in human p97 have been linked to diseases of aging and neurodegeneration, including IBMPFD (inclusion body myopathy with Paget’s disease of bone and frontotemporal dementia) and familial amyotrophic lateral sclerosis (ALS) [14,15,16,17].

p97 comprises an N-terminal domain (NTD), two ATPase domains, D1 and D2, containing conserved Walker A and B motifs, and an unstructured C-terminal tail [2]. The NTD and C-terminal tail are the key domains regulating ATPase function [10,13,18,19]. Six p97 monomers form a functional complex, featuring a double-ring structure based on the association of D1 and D2. The D1 domain is responsible for p97 oligomerization, regardless of their nucleotide-binding states. Upon ATP binding to the D1 domain [20,21,22], the D2 domain is the primary mediator of ATP hydrolysis [21,23,24,25,26,27]. The two stacked rings create a central channel in the p97 hexamer that binds substrates, facilitating the segregation of substrates destined for proteolysis from their usual location in multi-subunit complexes, organellar membranes, and chromatins [28,29]. Previous cryogenic electron microscopic (cryo-EM) structures of the AAA+ ATPases revealed that the substrate processing by the Cdc48 ATPase results in a broken ring and an asymmetric structure of the six p97 monomers. These structures have been used to suggest a hand-over-hand model for the substrate’s induced motion [30,31,32]. The substrate movement along the central channel is dependent on the D1 nucleotide state and is mainly driven by the ATP hydrolysis on the D2 domain [33,34]. This movement would call for cooperation between domains within the AAA+ ATPase complex and cofactor bindings [10,35,36]. However, it is not certain whether the hand-over-hand model applies to other p97 functions, such as membrane fusion.

The p97 NTD mediates the binding of various cofactors and ubiquitylated substrates and modulates ATPase activity [13,19]. Although the p97 NTD does not directly relate to nucleotide binding, it has two distinct configurations that are affected by the D1 nucleotide-binding states: the ‘up’ (ATP-bound D1) configuration, rising above the plane of the D1 ring, and the ‘down’ (ADP-bound D1) configuration, coplanar with the D1 ring [26,27,37]. However, previous EM structural analysis revealed variant NTD positions, rather than the only two distinct states [19,38]. One hypothesis suggests that the NTD configuration affects linker conformation, initiating inter-subunit or intra-domain communication, thus influencing ATPase activity [39,40,41]. This is supported by the position of IBMPFD-associated disease mutants that result in elevated ATPase activities, mostly from the D2 ATPase [19,42]. The IBMPFD mutations cluster on the NTD or the N-D1 linker of the p97 [26,43]. Because the mutation sites are far away from the D2 nucleotide-binding site, it has been suggested that the NTD mutation changes the interactions between p97 inter-domains, leading to functional alterations [34]. However, the connection between the pathological effects of the IBMPFD mutations and elevated p97 ATPase activity remains unclear.

In addition to mutations, the binding of p97 cofactors on the NTDs also impacts ATPase activities. p47 is one of the p97 cofactors involved in p97-mediated membrane fusion [7]. It contains an N-terminal UBA (ubiquitin-associated) domain, followed by a SEP (shp1, eyc, and p47) and UBX (ubiquitin-regulatory X) domain, which interacts with the p97 NTD (Figure 1A) [24,40,44]. The interaction reduces the D1 ATPase activities of both wild-type and p97 mutants [45]. Also, p47 enhances the D2 ATPase activity of wild-type p97 (p97^WT^), whereas it reduces the activities of the p97 disease mutants [45]. Thus, p47 has different effects on the two ATPase domains of both p97^WT^ and its mutants, resulting in that the mutant p97-p47 complex blocks the ATP hydrolysis in one single sigmoidal phase, rather than a biphasic response of the wild-type p97-p47 complex [45].

The binding mode of the p47 UBX domain to the p97 NTD is conserved among the UBX or UBX-like (UBX-L) containing cofactors [46]. Previous structures of partial or complete complexes reveal this interaction as seen in p97^N-D1^-p47^UBX^, p97^N^-FAF1^UBX^, p97^N^-OTU1^UBXL^, and p97-ASPL [24,47,48,49,50,51]. The high-resolution crystal structure of the p97^N-D1^-p47^UBX^ truncated complex showed that a conserved β sheet with an S3/S4 loop in the UBX domain binds to a hydrophobic pocket formed by the two p97 N-subdomains (comprised of an N_a_ (four-stranded β barrel) and an N_b_ (ψ barrel) domain) [24]. Recently, the bipartite and tripartite binding modes have been proposed for the interactions between multiple p47 and wild type p97 NTDs [51]. However, how this binding in the combination of disease NTD mutations modulates the D1 and D2 ATPase activities remain uncertain. Most of the current data for p97 mutants were based on the structures of truncated domains [52]. Because the structural information of the full-length diseased complexes is still lacking, more efforts on studying their functions and structures are needed to understand the mechanistic details.

Here we performed functional and structural analyses on the disease-linked complex, p97^R155H^-p47, to understand how the cofactor binding and NTD mutation leads to functional alterations. We chose p97^R155H^ as our target, since the NTD mutation of arginine 155 to histidine (R155H) is the most commonly found in the IBMPFD phenotype [53]. We have previously identified that p97 disease mutants are defective in the cofactor-regulated ATPase hydrolysis cycle [45]. Also, the increased p47 expression can correct the autophagy defect caused by the p97 disease mutant [45]. To understand how p47 impacts the function of the p97^R155H^ disease mutant, we characterized the interactions between the p97^R155H^ and p47 using single-particle cryo-EM image analysis. The structural results revealed two major forms in the molecular population: the p97^R155H^ dodecamer and the p97^R155H^ hexamer-p47 complex. The nucleotide-binding pockets are empty in the p97^R155H^ dodecamer. The presence of the nucleotides, such as ADP or ATP*γ*S, destabilizes the dodecameric formation. The NTDs showed all up configurations when bound to ATP*γ*S or in the absence of nucleotides. However, in the presence of ADP, the NTDs of the mutant allow variable intermediate positions, which may affect the accessibility of the p47 cofactor. Our structures also showed that the arginine fingers exhibit different conformations than those of the wild type when p97^R155H^ binds to p47. We propose a possible model that impacts functional alteration of the p97^R155H^ upon p47 binding, providing an insight into the pathological function of the IBMPFD-associated p97 mutants.

## 2. Results

### 2.1. Two Molecular Populations Are in the p97^R155H^-p47 Assembly

To analyze the activity of mutated p97, we first assembled an in vitro complex of full-length p97^R155H^ with p47 in the absence of nucleotides (molar ratio of p97^R155H^ hexamer:p47 monomer 1:10). A peak representing the p97^R155H^-p47 complex, corresponding to a molecular weight of about 670 kDa, was detected in the size-exclusion chromatographic (SEC) profile (Figure 1B). The peak profile showed a leading shoulder, implying that multiple molecular species were present in the peak fraction. We further characterized the peak fractions using SDS-PAGE, which confirmed the presence of both p97^R155H^ and p47 proteins in the main peak fraction and only the p97^R155H^ proteins in the leading shoulder (Figure 1C). This observation corroborates the previous report on the IBMPFD mutants, which showed that a full-length p97^R155H^ dodecamer was present in the leading fraction [19]. In contrast, the p97^WT^-p47 assembly did not show an asymmetric leading peak profile (Figure 1B). Thus, the p97^R155H^ mutant is found in additional forms, forming complexes beyond the hexameric arrangement of the wild-type protein. Although the higher-order form has been reported for the wild type [19], inhibitor-bound [54,55], or engineered p97 [56], we did not find the same pattern in the SEC of the wild type, possibly because the high-order complex was in a less proportion in the wild type that could not be detected. In the SDS-PAGE analysis, these higher-order complexes do not associate with p47, even in the presence of excess p47 (Figure 1B). This suggests that the higher-order arrangement of p97^R155H^ may hinder the interaction with the p47 cofactor.

If the high order p97^R155H^ does not interact with p47 and forms in a larger proportion than the wild type, the p47 binding affinity to the p97^R155H^ on average is lower than that of the wild type p97^WT^. To test this idea, we further employed fluorescent labeling and measured the temperature-related intensity change (TRIC) signal to determine the affinities of the p47 binding on p97^WT^ and p97^R155H^ and their dissociation constants, *K_d_*. In the absence of nucleotides, the resulting *K_d_* averages of p97^WT^-p47 and p97^R155H^-p47 are 80 nM and 132 nM, respectively, suggesting a slightly higher binding affinity for the wild type (Figure 1D).

### 2.2. Up NTDs of the p97^R155H^ Mutant and p47 Are Structurally Disordered

To visualize the interaction between the p97^R155H^ mutant and the p47 cofactor, we plunge-froze the sample of the p97^R155H^-p47 assembly for single-particle cryo-EM analysis. Notably, two different particle populations were discernible in the two-dimensional (2D) class averages (Appendix A). One showed a typical side view of the p97 hexameric form, and the other showed two stacked p97 hexameric double rings (Appendix A). Ab initio map generation produced two different three-dimensional (3D) densities presenting one with general p97 hexameric features and the other containing a dimer of hexamers (Appendix A). We reconstructed the 3D density maps of the p97^R155H^ dodecamer complex at 3.34 Å resolution and the p97^R155H^-p47 complex at 4.30 Å (Figure 2 and Appendix A). The ratio of the single-particle images of the p97^R155H^-p47 complex to the p97^R155H^ dodecamer is 1.56:1.

Local resolution estimation showed that the density of the p97^R155H^ dodecamer or the p97^R155H^-p47 complex is well resolved in D1 and D2 domains but not the NTDs (Appendix A). For p97^R155H^ dodecamer, we did not find densities that could be assigned to p47. Based on specific map features of the p97^R155H^-p47 complex, we were able to identify the NTD and p47^UBX^ domains at a lower contour level (2.7σ) by docking available atomic models (PDB code: 1S3S) into the density map (Figure 2B) [24]. Four NTD densities could be assigned and shown in up configurations, and they are rendered into slightly different orientations in the azimuthal direction along the C6 pseudo-symmetrical axis (Figure 2B). Only one p47^UBX^ domain was found to fit into the density attached to the up NTD (Figure 2B). We did not identify any high-order form of the p47 bound to p97, and it is consistent with the previous finding that the p97 NTDs interfere with p47 oligomerization [51,57]. In addition, the flattened densities of the NTD of p97 and p47 are possible due to their dynamic and partially disordered nature, leading to the weak scattering signals and limiting resolution of these components (Figure 2B and Appendix A). We also performed image classification in 3D, but the result did not yield high-resolution densities for modeling the SEP or UBA domain of the p47 (Appendix A). This observation is also corroborated by a neural network data analysis [58], showing high variations in the NTD and p47 densities (Figure 2C). The densities of D1 and D2 rings were invariable across the reconstructions. Unlike the p97^WT^-p47 complex [51], the mutant NTD may be too disordered or mobile to present a structural feature with the p47 factor to be resolved by cryo-EM.

In this complex assembling, we did not supply any nucleotides in the purification steps, and nucleotide was not detectable in the D1 and D2 nucleotide-binding sites of the p97^R155H^ dodecamer or the p97^R155H^-p47 cryo-EM density maps. This observation differs from the previous study where ADP is always observed in the wild type p97 D1 ATPase, even when the nucleotide was not supplied in the sample buffers [23,26,59,60,61]. It is the first time that the full-length p97^R155H^ disease mutant structure is revealed by single-particle cryo-EM. One possibility could be that the nucleotide binding for the p97^R155H^ mutant is weak in the extremely low concentrations of nucleotides, so the densities of the nucleotides were insufficient to be detected in the averages of the cryo-EM images. Since the previous analyses were conducted using p97^WT^ or truncated p97 mutants, it is likely that the R155H mutation or the domain-domain interactions in the full-length p97^R155H^ reduces the affinity of the D1 domain for ADP [45,62]. Furthermore, because the ADP off-rate of the p97^R155H^ D1 ATPase is two-fold faster than that of the p97^WT^ D1 ATPase and the nucleotide binding in the D2 domain depends on the D1 nucleotide state, it indicates a high likelihood of the ADP binding instability in D1 ATPase of the full-length p97^R155H^ [62]. Apart from this, the previous SPR experiments showed that the R155H mutation reduces the ADP-binding affinity to p97 [61,63]. Thus, the p97^R155H^ mutant is likely to be structurally stable when the nucleotide-binding pockets are empty.

### 2.3. P97^R155H^ Dodecamer Is Stabilized by the Two Oppositely Stacked D2 Rings

Our SEC profile and SDS-PAGE analysis showed that the dimeric p97^R155H^ hexamers do not bind p47 (Figure 1B,C). To elucidate how the two hexamers organize into a higher-order complex, we built an atomic model along with the cryo-EM density map of the p97^R155H^ dodecamer (Appendix A). The two p97^R155H^ hexamers are oppositely packed against their D2 rings with a D6 symmetrical arrangement (Figure 2A and Figure 3A). We did not find any density assigned for the p47 cofactor, as corroborated with our biochemical characterization and the *K_d_* determination of the p97-p47 binding (Figure 1B–D and Figure 2A). Thus, this suggests that the p47 cofactor does not access the p97^R155H^ dodecamer.

The C-terminal tail of the p97 (residues Q764 to G806) that interacts with the VCP cofactors was previously reported as flexible and has not previously been structurally resolved for a full-length p97^R155H^ mutant [19,64,65]. The crystal structure of the C-terminus up to P774 of an engineered p97 variant has also been determined, but at a weak signal level [56]. This is because in our p97^R155H^ dodecamer, the C-terminal tail is sandwiched by the two hexameric D2 rings, limiting its mobility and allowing visualization of high-resolution details (Figure 3A). Our cryo-EM density map of the p97^R155H^ dodecamer was able to show an apparent density of the partial C-terminal tail (residues Q764 to F773) (Figure 3A). The C-terminal tail points away from the pore center, interacting with another C-terminal tail of the opposite neighbor to stabilize the packing of the two p97^R155H^ hexamers (Figure 3B). Packing occurs via largely polar-polar interactions between the D2 α helices and a hydrogen-bonding network of the C-terminal residues (D749, E756, Q760, Q764, and R766) (Figure 3B). π-stacking forces between the aromatic side chains of the C-terminal tails and the D2 α helices (F674, F682, F768, F771 and F773) also contribute to the packing of the two p97^R155H^ hexamers.

### 2.4. Nucleotide Binding Destabilizes the p97^R155H^ Dodecameric Formation

We did not find nucleotide binding in the p97^R155H^ dodecamer structure, which occupies about 39.1% of the population (Figure 1B,C and Appendix A). In contrast, p97^WT^ is always associated with ADP in the D1 ATPase and has a trace amount of dodecameric form [19]. This could be possible because the nucleotide binding affects p97 oligomerization. We would like to test whether the nucleotide in the solution affects the dodecameric formation. We then mutated p97^R155H^ on the Walker A motif of the D1 (K251R) or D2 (K524R) ATPase to disrupt the nucleotide binding. The Walker A mutants do not bind any nucleotides nor perform the function. These double mutants were then imaged using negative-stain electron microscopy (EM) for single-particle image analysis. 2D class averages of these mutants showed that the ratios of the dodecamer to hexamer are 6.94:1 and 6.69:1 for p97^R155H-K251R^ and p97^R155H-K524R^, respectively (Appendix A). Compared to the above cryo-EM analysis of the p97^R155H^-p47 assembling, the ratios of the p97^R155H^ dodecamer in D1 and D2 Walker A mutants increase 10.8 and 10.4 times, respectively. This suggests that the p97^R155H^ dodecamer is stable when it does not bind nucleotides. Moreover, the double mutants lack ATPase activity, suggesting that the dodecameric p97^R155H^ is likely to represent an inactive state.

### 2.5. P97^R155H^ Dodecamer Is Likely to Be an Inactive Form

To compare the monomeric structure of the p97^R155H^ dodecamer with others, we superimposed our structure with p97^WT^ in ADP or ATP*γ*S bound state and a CB-5083-bound p97 (PDB codes: 5FTK (ADP), 5FTN (ATP*γ*S), and 6MCK (CB-5083)) (Figure 3C) [37,54]. The superposition of p97^R155H^ and p97|_ATP_*_γ_*_S_ revealed a downward rigid-body movement of the D2 domain, which may be induced by the torsional change of the D1-D2 linker (Figure 3C). Another significant structural change occurs in the C-terminal tail conformation. The C-terminal tail of the p97|_ATP_*_γ_*_S_ points to the pore center, which is in the opposite direction, against that of the p97^R155H^ dodecamer (Figure 3C). R766 on the C-terminal tail has been found to interact with the *γ*-phosphate of the D2-bound nucleotide directly [56]. Because the C-terminal tail of the p97^R155H^ dodecamer extends outwards from the pore center and away from the D2 nucleotide-binding site, it is impossible to stabilize the nucleotide binding. The D1 helix-turn-helix (HTH) motif (410–445) of the p97^R155H^ dodecamer differs from that of the p97|_ADP_, where the HTH motif interacts with the NTD and stabilizes it in the down configuration (Figure 3C). However, the HTH motif in the p97^R155H^ seems to not interact with the NTD, allowing NTD to be less spatially restricted in the up configuration. Although the NTD densities were resolved in low resolutions partly due to its mobility, the NTD densities in low χontours (2.4σ) could still be identified in the up configuration (Appendix A).

CB-5083 is a highly selective D2 ATP-competitive inhibitor on p97 ATPase, and the packing of the dimeric hexamers was also found in the crystal structure of the NTD-truncated p97 with CB-5083 [54]. Based on these superpositions, the structure of the p97^R155H^ dodecamer more closely resembles p97|_CB-5083_ (RMSD 0.747 Å) than p97|_ADP_ (RMSD 0.919 Å) or p97|_ATP_*_γ_*_S_ (RMSD 1.127 Å) (Figure 3C). The crystal structure is consistent with our dodecamer structure, suggesting that the p97^R155H^ dodecamer could be functionally equivalent to the inhibited p97 with CB-5083 (Figure 3C) [54].

### 2.6. Nucleotide Binding Influences the p47 Binding onto p97^R155H^

The nucleotide-bound state of the p97^WT^ D1 domain affects the D2 ATPase activity and influences the binding of the p47 cofactor [2,62]. In addition, p47 has a differential influence on the D1 and D2 ATPase functions of wild-type and p97 mutants [45]. To uncover how nucleotide binding modulates p47 binding to the p97^R155H^ mutant, we assembled the complexes in the presence of ADP or ATP*γ*S and biochemically analyzed the resulting complexes. ATP*γ*S was used to mimic the ATP-bound site since it cannot be hydrolyzed when bound to the nucleotide-binding site. The SEC profiles of the two assemblies in the presence of ADP (p97^R155H^|_ADP_-p47) and ATP*γ*S (p97^R155H^|_ATP_*_γ_*_S_-p47) were similar, and the peaks suggested a uniform distribution for both complexes (Appendix A). SDS-PAGE gel analyses verified the peak fractions containing p97^R155H^ and p47 (Appendix A).

We then collected the peak fractions and froze the samples for cryo-EM structural analysis (Appendix A). Although no leading fraction was observed in the SEC profile of the p97^R155H^|_ADP_-p47, a small fraction of the p97^R155H^ dodecamers were identified in the 2D image class averages (Appendix A). After 3D classification, the number of particle images representing the p97^R155H^ dodecamer was only 2219 (1.37%) among an overall 162,269 particle images. Like the 3D reconstruction of the p97^R155H^ without nucleotides, the dodecamer does not show any sign of bound nucleotide or p47 cofactor. However, the p97^R155H^|_ATP_*_γ_*_S_-p47 image dataset did not have the p97^R155H^ dodecameric population, suggesting that p97^R155H^ dodecamer does not form in the presence of ATP*γ*S (Appendix A). Thus, these suggest that the nucleotide presence, particularly ATP*γ*S, favors the dissociation of the p97^R155H^ dodecamer into hexamers. Furthermore, because the ATPase activity of the p97^R155H^ is higher than p97^WT^ [3,19,42,45,61,66] but the p97^R155H^ dodecamer does not interact with nucleotides, it is very likely that the increased activity of the p97^R155H^ is contributed from the functional hexamers, rather than dodecamers.

We calculated the 3D reconstructions of the p97^R155H^|_ADP_-p47 and p97^R155H^|_ATP_*_γ_*_S_-p47 complexes at 4.50 Å and 4.23 Å resolution, respectively (Appendix A). Cryo-EM densities of the nucleotide can be identified in both the D1 and D2 nucleotide-binding pockets of the p97^R155H^|_ATP_*_γ_*_S_-p47 density (Appendix A). Unlike the p97^R155H^-p47 and p97^R155H^|_ATP_*_γ_*_S_-p47 densities, the p97^R155H^|_ADP_-p47 density map showed a heterogeneous distribution of NTD orientations (Appendix A). An NTD density with bound p47^UBX^ domain was identified (1.0σ), located at the highest height among all the other NTDs (Appendix A). The tilting angles and heights of the NTDs relative to the ring plane varied sequentially (Appendix A). This observation of the high structural variabilities is consistent with the deep coordinate neural network analysis (Appendix A).

For the p97^R155H^|_ATP_*_γ_*_S_-p47 complex, four NTDs and two p47^UBX^ structures can be assigned to the cryo-EM density, and the unassigned two NTD densities were fragmented (Figure 4A). The four up NTD are consistent with the crystal structure of the truncated N-D1 domain of the p97^R155H^|_ATP_*_γ_*_S_ [26]. Like the p97^R155H^|_ADP_-p47 complex, the densities of NTDs and p47 were highly variable, but those for D1 and D2 domains were almost invariant, as shown in the local resolution estimation (Appendix A).

### 2.7. P47 Binding Impacts p97^R155H^ Function via an Allosteric Effect on ATPases

Previous functional analyses demonstrated that cofactor binding or NTD mutation alters p97 ATPase function, mainly the D2 ATPase [42,45]. However, the NTD is far away from the nucleotide-binding pocket and how do NTDs allosterically regulate p97 ATPase functions? Because D1 and D2 ATPase activities are correlated, one possibility is that the NTD regulates ATPase functions via inter-domain or interprotomer communications between these functional modules [42,67]. Because the cryo-EM density of p97^R155H^|_ATP_*_γ_*_S_-p47 allows us to visualize the side chains in its D1 and D2 domains, we compared it with the model with p97^WT^|_ATP_*_γ_*_S_ structure (PDB code: 5FTN) [37]. We identified two p47_UBX_ bound onto the p97^R155H^ NTDs, implying majority of the p97^R155H^ particles bind two p47 molecules. The superposition of the structures of the p47-bound p97^R155H^|_ATP_*_γ_*_S_ and p97^WT^|_ATP_*_γ_*_S_ alone showed that the D2 domain slightly moves and most of the critical residues in the nucleotide-binding pockets do not exhibit major conformational changes (RMSD: 1.015 Å) (Figure 4B). However, the D2 R635, the D2 arginine finger, of the p97^R155H^|_ATP_*_γ_*_S_ changes its conformation upon p47 binding, although the conformations of the D1 arginine fingers seem unchanged (Figure 4B). It is also known that the arginine fingers extending from the neighbor protomer can affect the ATP hydrolysis efficiency [3,68,69].

We would then like to test whether the D2 ATPase activity change is affected by NTD mutation or p47 binding via the arginine fingers. We introduced two mutations of the arginine fingers on R359A and R635A and measured the mutant ATPase activities (Figure 4C). The p97^R359A^ and p97^R635A^ mutants have reduced activities compared to the wild type (59% and 5%, respectively), and the mutation of the D2 R635 nearly abolished the p97 ATPase activity, consistent with the previous findings [70]. We next prepared the two double mutants, p97^R155H-R359A^ and p97^R155H-R635A^, and the results showed that both double mutants reduced p97^R155H^ activities to 26% and 2.3%, respectively. Because the p97^R155H^ has a higher ATPase activity than p97^WT^ (Figure 4C) [3,19,42,45,61,66], the results showed that the increased p97^R155H^ ATPase activity due to its R155H mutation was diminished by the D1 and D2 arginine mutations. Both of the D1 and D2 arginine fingers appear to affect the communications within p97 subunits for the R155H functional enhancement. Thus, the gained ATPase activity from the R155H mutation of the NTD is very likely to associate with functional arginine fingers.

Next, we would like to know whether p47 impacts p97^R155H^ function via arginine fingers in the same manner as R155H mutation. As seen previously, upon binding to p47, p97^WT^ exhibits a biphasic response, but p97^R155H^ does not exhibit the rebound Phase 2 (Figure 4D) [45]. However, the D1 arginine finger mutants, p97^R359A^, lacked Phase 1 inhibition but showed significant activation in the later phase (1.9-fold), similar to the D1 Walker B mutant (E305Q), which binds to nucleotide but does not perform the catalysis (Figure 4D) [45]. Similar to p97^R359A^, the D2 arginine finger mutants, p97^R635A^, showed a significant Phase 2 activation (3-fold), but different from the D2 Walker B mutant (E578Q) (Figure 4E) [45]. These data indicate that upon p47 binding, the Phase 1 inhibition of the p97^WT^ associates with both D1 and D2 arginine fingers, but the Phase 2 activation is mainly contributed by the D2 ATPase activity that associates with the D2 arginine finger (Figure 4D,E). Thus, p47 likely impacts both wild type and p97^R155H^ mutant via arginine fingers, particular in Phase 2.

The responses of the double mutants showed both impacts from R155H mutation and p47 binding. The p47 inhibitory regulation on the p97^R155H^ was abolished for the two double mutants, the p97^R155H-R359A^ and p97^R155H-R635A^ mutants (Figure 4C,D). Especially for the p97^R155H-R635A^ mutant for the D2 arginine finger, the curve is flat, implying that the p47 binding has no impact on the ATP hydrolysis. Thus, this may indicate a functional connection between p47 binding and the arginine fingers, R359 and R635, of the p97^R155H^ mutant. The structural superpositions showed that the change of the ATPase function may result from the slight conformational changes of the arginine fingers (Figure 4A). Thus, these findings suggest that the arginine fingers are critical for the p47-induced communications between the NTD and ATPase domain.

In addition to the functional modulation of p47 via arginine fingers, we would like to test whether the p47 binding is affected when the arginine finger is mutated. We fractionated a mixture of p97^R155H-R359A^ or p97^R155H-R635A^ (1.67 μM in hexameric form) and p47 (80 μM in monomeric form) proteins in a gel-filtration column, using p97^R155H^ as a positive control (Appendix A). We found that the double mutant p97^R155H-R359A^ and p97^R155H-R635A^ retained the ability to bind p47, eluting in the same fraction as the p97^R155H^-p47 complex (Appendix A). We also determined the binding affinities, *K_d_*, of the p47 to p97^R155H^, p97^R155H-R359A^, and p97^R155H-R635A^ as 132, 179, and 228 nM, respectively (Appendix A). Thus, these results showed that the gained ATPase activities of the p97^R155H-R635A^ from p97^R155H^ impacted by p47 may be partly due to weak p47 binding affinity to the p97^R155H-R635A^ double mutant (Figure 1 and Figure 4E and Appendix A). In addition, the impaired arginine fingers reduce the p47 binding affinity to p97^R155H^, implying that the domain-domain communications within p97 ATPase affect how p97 responds to cofactor binding.

## 3. Discussion

Single amino acid mutations in p97 have long been linked to diseases, including IBMPFD and ALS [71], and these disease mutants alter the p97 ATPase activity and cofactor binding on the NTDs [45]. Because the NTD is far away from the D2 ATPase, where ATP hydrolysis mostly occurs, it is unclear how the NTD allosterically affects the p97 ATPase function. The molecular mechanism underlying the disease is still unresolved today. Our goal aims to understand (1) how NTD conformational change, such as up and down configurations, connects to ATPase function, (2) the structural change of p97 ATPase caused by R155H mutation, and (3) how the mutant responds to p47 binding in the context of a complete full-length complex. Here we conducted biochemical and structural analyses to study potential pathological changes in the p97^R155H^ mutant, the most prevalent disease-linked mutation [53]. To our knowledge, our report reveals the first full-length p97^R155H^ mutant structures in dodecameric and hexameric forms, as well as the p97^R155H^-p47 complex in different nucleotide states. Previous studies have placed less emphasis on the p97^R155H^ dodecamer. Our report found that the p97^R155H^ dodecamer seems unstable in the presence of nucleotides and insensitive to bind p47. Although we do not know if the p97^R155H^ dodecamer plays a physiological role in vivo or leads to IBMPFD disease, our cryo-EM image analysis showed that this high order form can be stabilized in an aqueous solution and occupies about 40%. We also found a possible connection for the p47 and NTD regulation on ATPase function, likely arginine fingers, and the miscommunication between these domains alters the normal ATPase activity. This may interfere with the ability of the p97 to carry out the p47-related function such as Golgi-membrane reassembly or autophagy [30]. These new findings thus reveal how cellular functions may be so profoundly impacted in the p97^R155H^ mutant.

The formation of a p97^WT^ dodecamer has been previously reported [47,56,72]. However, our SEC analysis did not show the dodecameric fraction in the wild-type assembly, and the proportion of the p97^WT^ dodecamer may be too low to be detected in our SEC analysis (Figure 1B). Although it is not known whether the formation of the p97^R155H^ dodecamer occurs in vivo or whether it plays any role in a cell, the dodecamer at least can be stably formed in the solution and likely in the cell as well. In addition, the structural comparison of the CB-5083-bound structure [54] and its empty nucleotide-binding pockets highly suggested the dodecameric form as an inactive state. The inactivation of the p97 dodecamer could be caused by its high-order organization, which may hinder the mobility of the monomers, interfere with the hexameric ring to break, prevent hand-over-hand movements and possibly, in turn, down-regulate normal p97 ATPase function during substrate processing [31,32]. Only when the nucleotides are present, the p97^R155H^ dodecamer will dissociate into two functional hexamers with an elevated ATPase activity. However, we have no information about why the R155H mutation favors the dodecameric formation. One possible explanation would be that the dodecameric formation induced by R155H mutation may keep p97 inactive while under the stress of the limited ATP concentrations in the cell [54]. The definition of its functional role in vivo requires further investigations using cell biological tools.

Our data showed that the p47 binding or NTD mutation affects the communications between the functional modules of the full-length p97^R155H^ ATPase in either an intra-domain or an inter-domain manner or both. In addition, the affinity of p47 to p97^R155H^ is affected by the D1 or D2 ATPase activities. We also showed that either D1 or D2 arginine finger mutation can affect the p47 binding affinity. These views might not be observed using the truncated or incomplete p97 complexes to show the effects caused by domain communications. The partial p97 ATPase may also lead to inconsistent observations on the stoichiometry of the p97-p47 bindings. For example, the crystal structure of the p97^N-D1^-p47^UBX^ showed that two p47 UBX domains bind to two adjacent p97 monomers, while a third is poorly resolved located away from those two bound NTDs [24]. However, this view is not consistent with our observation from single-particle cryo-EM. Therefore, a complete full-length complex will be required to provide a view of the interactions between overall functional modules that are close to their native states.

Previous structural studies have revealed the interactions between p97^WT^ and p47 proteins [24,51,57,73]. NMR (nuclear magnetic resonance) spectroscopic analysis of the p97^N-D1-L^-p47 (residues 1-480 of p97) indicated a possible tripartite p47 binding mode [51]. This interaction involves the UBX and the C-terminal SHP of p47 binding to one p97 NTD, while a third p47 N-terminal SHP domain binds the adjacent NTD of the p97. The up NTD configurations allow organizing p47 in this arrangement, rather than down NTDs. On the other hand, the p97^R155H^ mutant, especially when bound to ATP, should interact with p47 cofactors followed by the proposed tripartite binding mode, since the NTDs are all in up configuration [51,62]. However, our p97^R155H^-p47 complex structures indicate a disordered or flexible structure of NTD or p47^SHP^ domain, implying that p97 NTD mutant may either interact with p47 more transiently or require a concerted motion on NTDs to adapt the proposed binding mode [40,62]. In this sense, the oscillating up NTDs of the p97^R155H^|_ADP_ could easily access the p47 cofactors than the p97^WT^|_ADP_, which has all down NTDs [37]. This corroborates with the previous finding that the binding of p47 on the p97^WT^ ATPase in the presence of ADP is much weaker than those in the presence of ATP or the absence of nucleotides [62]. However, the p47 binding for the p97^R155H^ mutant is independent of the nucleotide states [62].

Recent cryo-EM structures of the AAA+ ATPase with p47 cofactor or its homolog showed an asymmetric arrangement of the hexameric structure [31,35]. These structures have shown the substrate binding in the central channel of the ATPase double ring [31,32,74]. However, we did not observe the same view in the structures of our complexes or p97^R155H^ ATPase alone. It is possible that the p47 binding alone or the R155H NTD mutation cannot induce the domain movements and the presence of the substrate seems to be required. Ufd1/Npl4 (UN) is one of the p97 cofactors that increase p97 unfoldase activity [75,76]. The p97 disease mutant exhibited an elevated unfoldase activity when binding to the UN cofactor and the substrate [38]. Although p47 seems not to be involved in the unfoldase process, it may regulate p97 ATPase in a similar manner for Golgi membrane reassembly or autophagy, which may not require substrate in the pathway. The p97-p47-mediated membrane fusion requires the participation of VCIP135 (valosin-containing protein p97/p47 complex-interacting protein p135) and syntaxin-5 (Syn5) [77], which help in the coordination of ubiquitin transfer and ATP hydrolysis. These cellular functions may utilize an action mode of p97 different from the ERAD (endoplasmic-reticulum-associated protein degradation) process or require more proteins to initiate the conformational changes to perform the function.

Here we propose a possible diseased mechanistic model for p97^R155H^ regulation (Figure 5). In the absence of nucleotides, the p97^R155H^ mutant prefers to form a higher-order p97^R155H^ dodecamer, immobilizing C-terminal tails and prohibiting interaction of p47 cofactor with the NTD. When the nucleotide is present and binds to the enzyme mutant, the p97^R155H^ dodecamer dissociates into two functional hexamers, allowing cofactor binding and performing ATP hydrolysis. The regulation of the p47 cofactor binding and functional alteration by R155H mutation may induce a conformational change of the arginine fingers and modulate the ATPase function. Our analyses suggest the importance of using a full-length, rather than truncated, p97 for functional or structural characterization to gain a complete view of the domain-domain communications within the p97 hexamer. To build on our finding regarding dysregulation of interprotomer communication in p97^R155H^ hexamer, we expect the incorporation of additional physiological substrates for p97-p47 assembly for the next critical step to understand the p97 pathological mechanism.

## 4. Materials and Methods

### 4.1. Overexpression of the Wild Type p97, p97R155H Disease Mutant, and p47 Proteins

Plasmid constructs used for generating p97 and its mutants and p47 proteins are listed in Appendix A. Overexpression and purification of the p97 and its mutants and p47 followed previous methods [61].

### 4.2. ATPase Activity Measurements

Detection of ATPase activity using Biomol Green reagent (Enzo Life Sciences, Farmingdale, NY, USA) was performed as previously described with slight modifications [45]. To compare the ATPase activity of wild type and mutant p97 proteins, each purified protein was diluted to a final monomer concentration of 25 nM in 50 μL ATPase assay buffer (50 mM Tris (pH 7.4), 20 mM MgCl_2_, 1 mM EDTA, 0.5 mM TCEP, 0.01% Triton X-100 and 80 nM BSA). After adding 200 μM ATP, the reaction was carried out at room temperature for optimal reaction times. 50 μL Biomol Green reagent (Enzo Life Sciences, Farmingdale, NY, USA) was added to stop the reaction and the absorbance at 635 nm was measured using a BioTek Synergy Neo 2 plate reader (BioTek, Winooski, VT, USA). The eleven-dose titrations with p47 cofactor were performed by adding the varying amount of p47 protein in ATPase buffer with p97. The results were calculated from five replicates using GraphPad Prism 7.0.

### 4.3. P47 Binding Affinity Measurements

The binding affinity (*K_d_* values) between p47 and p97^WT^ or p97 mutants was determined by measuring the temperature-related intensity change (TRIC) signals using a Dianthus NT.23 instrument (Nano-Temper Technologies, München, Germany). p47 was labeled with a RED-NHS dye using a Monolith Protein Labeling Kit RED-NHS second-generation (Nano-Temper Technologies, CAT# LO-L011). The full-length wild type or mutant p97 was titrated against 10 nM of p47-NHS in two-fold steps from 2.8 µM to 1.37 nM in 20 µL working buffer (20 mM HEPES (pH 7.4), 150 mM KCl, 1 mM MgCl_2_, 5% (*w*/*v*) glycerol, and 0.0025% (*v*/*v*) Tween 20). Assays were performed in a Dianthus 384-well plate and reproduced in three independent experiments. Data from the three independent measurements were fitted using non-linear regression analysis in Prism 7.0.

### 4.4. Assembling p97-p47 Complexes

To assemble p47 with p97^WT^ or p97 mutants, 1.67 µM p97 hexamer was mixed with 80 µM p47 for 10 min and fractionated in 20 mM HEPES (pH 7.4), 150 mM KCl, 1 mM MgCl_2_, and 5% glycerol. Three different assemblies were prepared in the absence of nucleotides and the presence of 100 µM ADP or ATP*γ*S in the SEC elution buffer. The mixture of the p97-p47 complex was loaded onto a Superdex 200 10/300 GL (GE Healthcare, Chicago, IL, USA) for size-exclusion chromatography (SEC). When the samples were used for cryo-EM analysis, 0.25% (*v*/*v*) glutaraldehyde (Electron Microscopy Sciences, Hatfield, PA, USA) was applied for on-column cross-linking by following the previous method [78]. Peak fractions were characterized using SDS-PAGE, Western blotting, and negative-stain electron microscopy (EM).

### 4.5. Negative-Stain Electron Microscopy for Single-Particle Analysis

The negatively stained specimens were prepared using the previous method [79]. 0.01 mg/mL of the protein sample was used to applied onto a continuous carbon-coated copper grid. The specimen was imaged using a Philips CM12 transmission electron microscope (TEM) (80 keV) with a side-mounted CCD camera (Model 791, Gatan, Pleasanton, CA, USA) or a FEI Tecnai TF20 TEM with a CCD camera.

For imaging the Walker A mutants, the specimens were imaged under a FEI Tecnai TF20 TEM, recording at a pixel size of 1.04 Å/pixel at the specimen level. 153 and 144 electron images were collected for p97^R155H-K251R^ and p97^R155H-K524R^, respectively, and imported into Relion (version 3.1-beta-commit-ca101f) [80] for image processing. 23,342 and 23,020 particles of p97^R155H-K251R^ and p97^R155H-K524R^ were manually selected from the electron images, respectively, and the two-dimensional (2D) class averages with an assigned *k* of 50 were calculated. For the p97^R155H-K251R^ dataset, the numbers of side views for dodecamer and hexamer were 2173 and 313, respectively. For the p97^R155H-K524R^ dataset, the numbers are 2007 and 300, respectively.

### 4.6. Cryo-EM Data Collection

A holey-carbon C-flat grid (2/1 4C; Protochips, Morrisville, NC, USA) was glow-discharged for 15 s using a Pelco easiGlow glow-discharge system (Ted Pella, Redding, CA, USA). A 6 µL protein sample was applied to the pretreated grid, and the excess solution was blotted using a filter paper (retention 20 µm) (Product #:47000-100, Ted Pella, Redding, CA, USA). The grid specimen was quickly plunge frozen into liquid ethane using a Thermo Fisher/FEI Vitrobot Mark IV automated freeze plunger (Thermo Fisher/FEI, Hillsborough, OR, USA) for 6 s in a chamber with a humidity of 100%. Particle homogeneity and ice backgrounds of the grid specimen were screened using a FEI Tecnai TF20 TEM. Grids with thin ice and a homogenous particle dispersion were used for subsequent cryo-EM data collection.

All the cryo-EM data in this report were collected at the Eyring Materials Center (EMC) at Arizona State University (ASU) (Tempe, AZ, USA) using a Thermo Fisher/FEI Titan Krios TEM (Thermo Fisher/FEI, Hillsborough, OR, USA) at an accelerating voltage of 300 keV. Cryo-EM movies were recorded using a Gatan K2 Summit direct electron detector (DED) camera (Gatan, Pleasanton, CA, USA). Defocus range was set to −0.8 to −2.5 µm. Nominal magnification was 48,077X, resulting in a physical pixel size of 1.04 Å/pixel at the specimen level. The movie data was recorded at a counted rate of 2 e^−^/sub-pixel/sec and a sub-frame rate of 200 msec in super-resolution mode. Total exposure was 6 s, accumulating to a dosage of 44.4 e^−^/A^2^. The beam-image shift was applied to accelerate data acquisition [81]. Data collection was automated using the customized SerialEM macros (version 3.9) [82].

### 4.7. Image Processing

For the p97^R155H^-p47 assembly, 4223 collected movies were unpacked and gain-normalized using Relion (version 3.1-beta-commit-ca101f) [80]. Image frames of a movie were translationally registered and averaged with a dose-weighting scheme. The final frame average was Fourier-cropped at the spatial frequency of 1.04 Å^−1^. The defocus and astigmatism of the images were estimated using the CTFFIND4 program (version 4.1.13) [83]. A few particles were manually selected from the images, and their average was then served as a searching template for the subsequent automated particle picking in Relion [80]. 368,575 selected particle images were automatically selected, and the data curation was completed using iterative 2D unsupervised classification. The classes with discernible features were selected for ab initio volume generation using cryoSPARC software (version 2.15) [84]. The two three-dimensional (3D) densities of a p97^R155H^ dodecamer and a p97^R155H^-p47 assembly were generated.

For particle images of the p97^R155H^ dodecamer, one additional round of 3D classification was performed to remove poorly aligned particle images. The selected 3D class average was then refined against 64,252 experimental particle images by enforcing a D6 symmetry. The density was further improved using CTF refinement (fit of the defocus and astigmatism per particle and estimation of beam tilt) [85] and Bayesian polishing [86] procedures to a final resolution of 3.34 Å. The final map was sharpened using a *b*-factor of −78.6 Å^2^.

For the p97^R155H^-p47 complex particle images, the consensus 3D volume of the p97^R155H^-p47 complex was calculated at 4.30 Å resolution with a sharpened *b*-factor of −125.7 Å^2^. The C6 symmetry was applied to further improve the densities of the D1 and D2 domains. The generated map reached a final resolution of 3.98 Å and was sharpened using a *b*-factor of −139.3 Å^2^. The final resolutions of all the density maps were determined using the golden FSC criteria at 0.143 cutoff [87], and the local resolution estimations were performed using the implementation in Relion software. The processing schematic is shown in Appendix A. On the other hand, further 3D classification procedures with *k* = 6 or larger or multiple hierarchical layers were attempted (Appendix A). However, no major structural differences between the generated class averages were discernible, and no improvements were shown in the density quality in local regions of NTDs and p47 (Appendix A).

For the p97^R155H^|_ADP_-p47 and p97^R155H^|_ATP_*_γ_*_S_-p47 datasets, image processing was generally conducted using cryoSPARC software (version 2.15) [84]. 3512 and 2796 movies of the p97^R155H^|_ADP_-p47 and p97^R155H^|_ATP_*_γ_*_S_-p47 were unpacked, gain-normalized and imported into cryoSPARC, respectively. Frame registration and averaging were performed using patch motion correction. Defocus and astigmatism parameters were estimated using patch CTF estimation. Particle locations were automatically selected using the Topaz program (version 0.2.3) [88]. For the p97^R155H^|_ADP_-p47 complex, 1,124,232 particle images were selected and curated using iterative 2D unsupervised image classification. The classes with discernible features were selected for ab initio volume generation. The two densities of a p97^R155H^ dodecamer and a p97^R155H^|_ADP_-p47 assembly were generated. 2219 and 160,050 particle images were used to calculate the final volumes of the p97^R155H^ dodecamer and the p97^R155H^|_ADP_-p47 complex, respectively. The two volumes were refined against their experimental particle images and reached resolutions of 6.10 Å (p97^R155H^ dodecamer) and 4.50 Å (p97^R155H^|_ADP_-p47). Like the p97^R155H^-p47 dataset, further 3D classification did not yield discernible features in the local regions for NTDs and p47. Processing schematic and local resolution estimation are shown in Appendix A. For the 97^R155H^|_ATP_*_γ_*_S_-p47 dataset, we reconstruct the volume following the same procedure as above. Final 63,353 particle images were used to reconstruct a 3D volume at 4.23 Å resolution, sharpened using a *b-*factor of −77.6 Å^2^. Processing schematic and local resolution estimation are shown in Appendix A.

Structural heterogeneity of the three p97^R155H^-p47 datasets was further analyzed using a deep coordinate neural network by the cryoDRGN program (version 0.3) [58]. Before latent encoding, the particle images were Fourier-cropped to a box size corresponding to pixel sizes of 3.90 and 3.71 Å/pixel for p97^R155H^-p47 and p97^R155H^|_ADP_-p47 dataset, respectively. For each group, an eight-dimensional latent variable model was trained for 25 epochs. The encoder and decoder architectures were three layers with 1024 nodes. The latent encodings were visualized using a principal component factor plot or UMAP (uniform manifold approximation and projection) representation [89]. After training, *k*-means clustering with *k* = 20 was conducted on the latent encodings, and reconstructions were calculated at the cluster centers using the decoder network.

### 4.8. Modeling

The previous p97 and p47 coordinates (PDB code: 5FTK and 1S3S) were used as the templates for atomic modeling or molecular docking [24,37]. The templates were first docked into individual cryo-EM densities using the ‘Fit in the Map’ function in UCSF Chimera (version 1.14) [90]. The fitted model was manually rebuilt using Coot (version 0.9.1) [91] and then refined against the cryo-EM densities using the ‘phenix.real_space_refine’ program in Phenix software package (version 1.18.2-3874) [92]. Hydrogen atoms were added using the ‘phenix.reduce’ program for the model refinement and removed after the refinement. Secondary structure restraints were applied during the model refinement. The refinement and validation statistics were listed in Appendix A. The figures for the cryo-EM density maps and atomic models were prepared using UCSF Chimera and ChimeraX (version 0.91) [93].

## Figures and Tables

**Figure 1 ijms-22-08079-f001:**
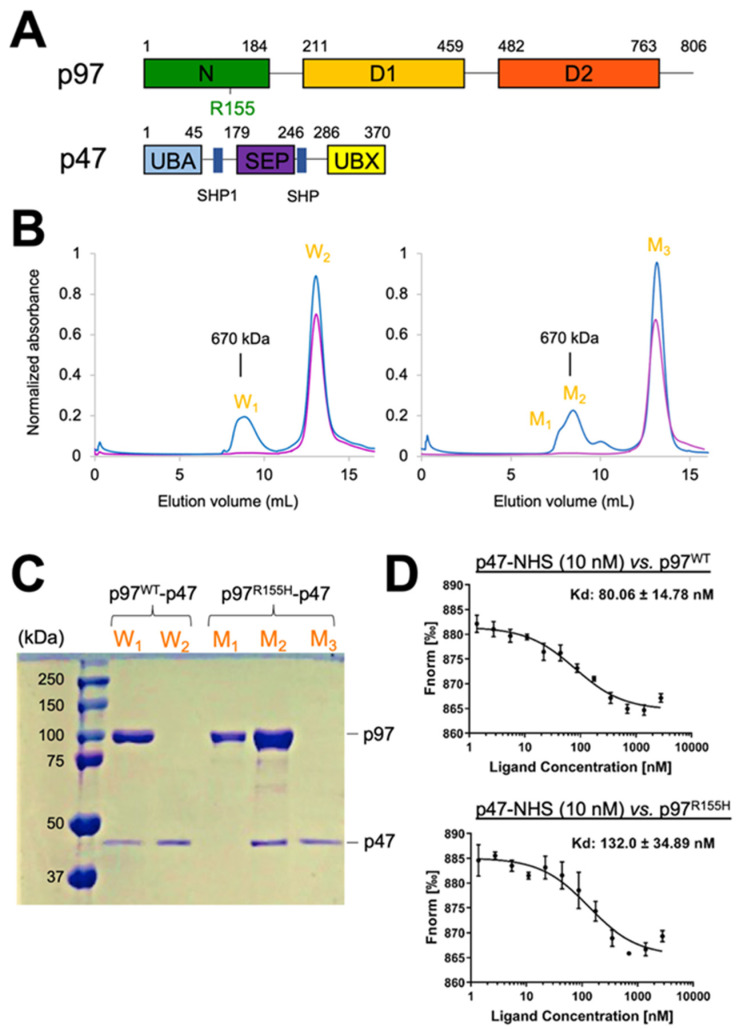
Characterization of the p97^R155H^-p47 and p97^WT^-p47 complex formation. (**A**) Domain structures of the p97 ATPase and its cofactor p47. The disease-linked mutation of R155H is labeled on the NTD of p97. (**B**) Size-exclusion chromatographic (SEC) profiles of the p97^WT^-p47 (left) and p97^R155H^-p47 (right) assemblies. Blue and purple curves are for the p97-p47 assembly and p47 protein alone. Peak fractions are labeled. (**C**) SDS-PAGE analysis of the SEC eluted peaks of the p97-p47 assemblies. (**D**) Determination of dissociation constants, *K_d_*, for the p97^WT^-p47 and p97^R155H^-p47 complexes. Unlabeled p97 was titrated against 10 nM of RED-NHS-labeled p47 in two-fold steps from 2.80 µM to 1.37 nM. The temperature-related intensity change (TRIC) signals were recorded and plotted against p47 concentrations.

**Figure 2 ijms-22-08079-f002:**
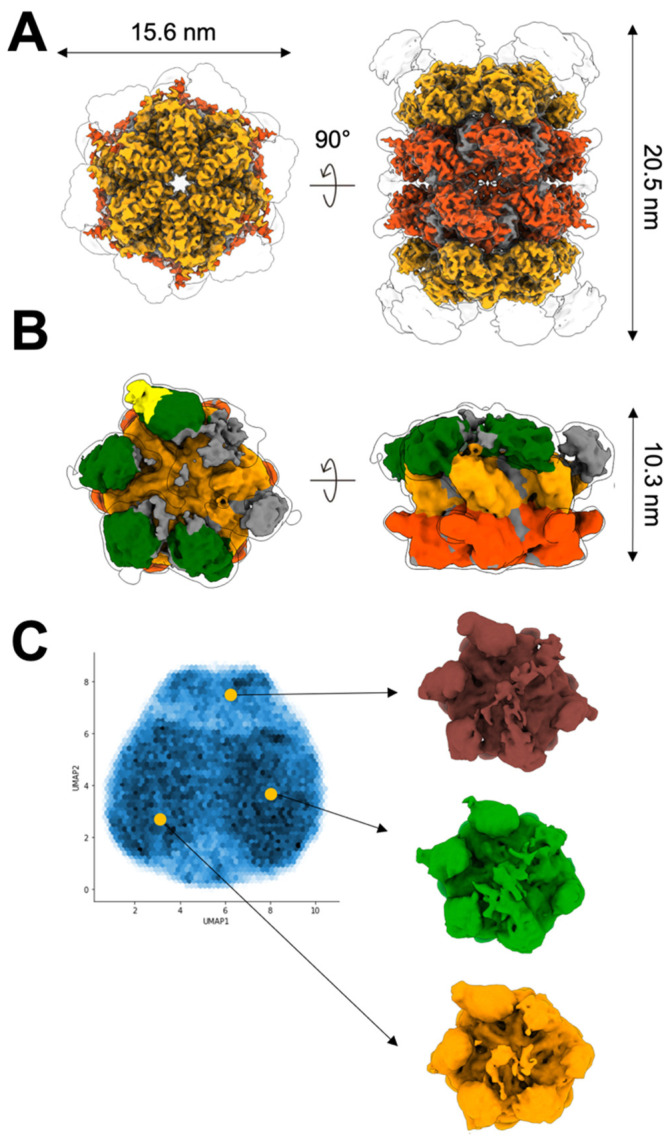
Cryo-EM reconstructions of the p97^R155H^ dodecamer and the p97^R155H^-p47 complex. Cryo-EM densities of (**A**) p97^R155H^ dodecamer (5.6σ) and (**B**) p97^R155H^-p47 complex (2.3σ). Green, orange, orange-red, and yellow highlight the NTD, D1, and D2 of the p97^R155H^ and p47^UBX^, respectively. Grey color indicates regions where densities could not be assigned. Envelopes are the cryo-EM maps at a lower contour (p97^R155H^ dodecamer: 2.4σ; p97^R155H^-p47: 1.6σ). (**C**) UMAP (uniform manifold approximation and projection) representation of a generative neural network analysis of the p97^R155H^-p47 cryo-EM density. Blue intensities represent the distribution of particle images in the embedded latent space. Three different 3D averages (brown, green and light orange) were calculated from the chosen clusters (orange circles).

**Figure 3 ijms-22-08079-f003:**
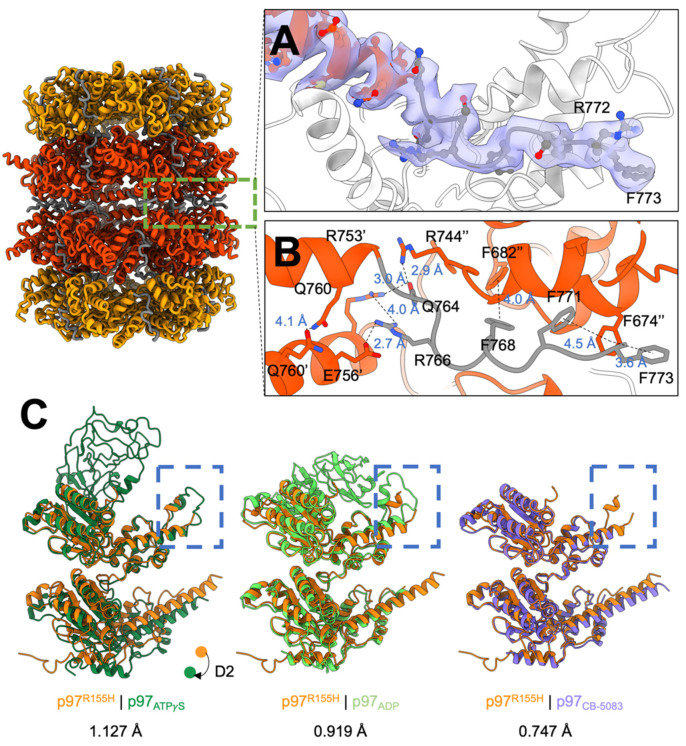
Cryo-EM structure of the p97^R155H^ dodecamer. Orange, orange-red, and grey indicate the D1, D2, and linker domains of the p97^R155H^. (**A**) Cryo-EM density of the partial C-terminal tail. The purple surface shows the cryo-EM density (4.6σ). The side chains are shown in stick representation. (**B**) Interaction network of the binding interface of the two p97^R155H^ hexamers. The two hexamers interact via hydrogen bonds and π-π aromatic packing. (**C**) Superposition of the p97^R155H^ with p97 structures in different nucleotide states. Dark green, light green, and purple are p97_ATP_*_γ_*_S_, p97_ADP_, and p97_CB-5083_, respectively. RMSDs are 1.127 Å, 0.919 Å, and 0.747 Å for p97_ATP_*_γ_*_S_, p97_ADP_, and p97_CB-5083_, respectively. Blue rectangles highlight conformational changes in the D1 helix-turn-helix motif. The direction of the D2 movement upon ATP*γ*S binding is shown in dots and an arrow.

**Figure 4 ijms-22-08079-f004:**
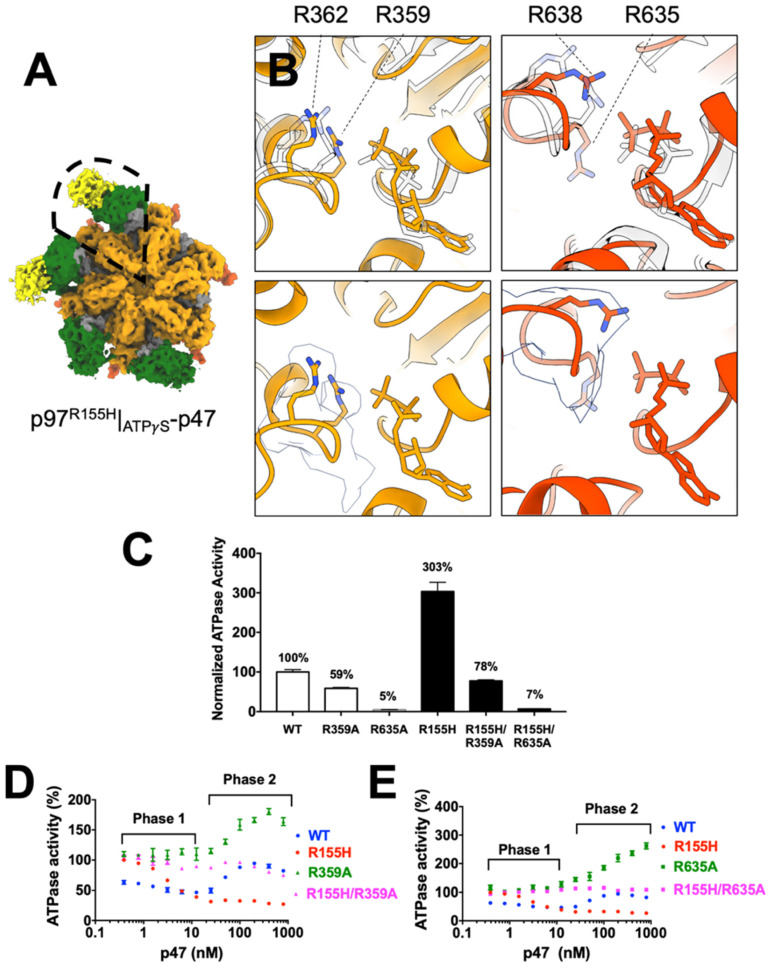
Cryo-EM structural analysis of the p97^R155H^|_ATP_*_γ_*_S_-p47 complex and functional measurements of the p97 arginine finger mutants. (**A**) Cryo-EM density of the p97^R155H^|_ATP_*_γ_*_S_-p47 complex. Color coding is the same as those in Figure 2. Enclosed is the p97^R155H^|_ATP_*_γ_*_S_-p47 monomer used for structural comparison with p97^WT^|_ATP_*_γ_*_S_. (**B**) Superposition of the structures of p97^R155H^|_ATP_*_γ_*_S_-p47 and p97^WT^|_ATP_*_γ_*_S_ (PDB code: 5FTN) [37]. Bound ATP*γ*S and arginine fingers are shown in stick representation. Upper panels are the superpositions of the two structures and white are the p97^WT^|_ATP_*_γ_*_S_ structure. Lower panels are the map-model fitting of the two arginine fingers in D1 and D2 nucleotide-binding sites. White surfaces are cryo-EM densities. (**C**) Full-length (FL) p97 ATPase activities of wild type and R155H mutants. Activity measurements were normalized relative to the p97^WT^ activity and measured in the presence of 200 µM ATP (*n* = 4). (**D**) Normalized ATPase activities of p97 ATPase: WT (blue circles), R155H (red circles), R359A (green triangles) and R155H-R359A (magenta triangles). The R359A mutation is located in the D1 arginine finger. p97 ATPase activity was normalized relative to its basal activity in the absence of p47. The addition of p47 increased from 0 to 800 nM. ATPase activity measurements were performed in the presence of 200 µM ATP. (**E**) Experiments as in (**B**), but the mutated residue was located at the R635A, the D2 arginine finger. p97 ATPase activity in R635A and R155H-R635A mutants is shown in green squares and magenta squares, respectively. All error bars indicate ±SD (*n* = 4).

**Figure 5 ijms-22-08079-f005:**
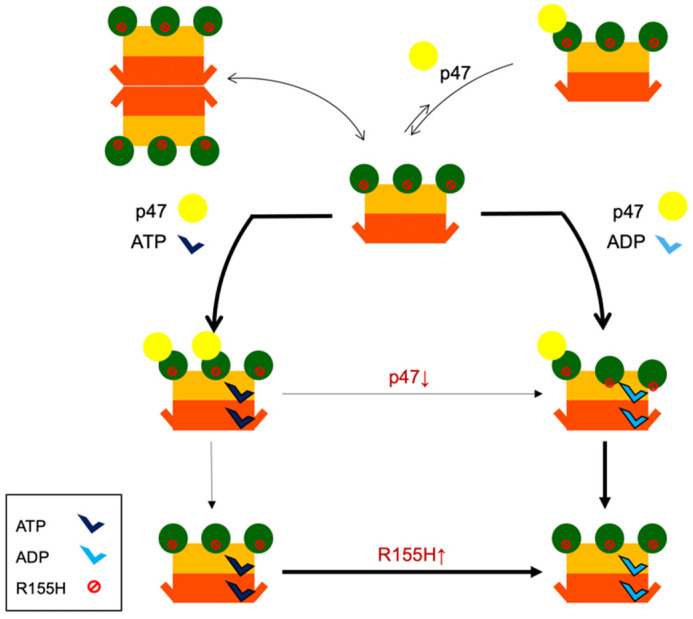
Proposed model for p47 binding to the disease-linked p97^R155H^ mutant. The NTD, D1, and D2 domains of the p97^R155H^ are colored green, orange, and orange-red. In the absence of nucleotides, p97^R155H^ can form a dodecamer and become inactive. p47 does not access the p97^R155H^ dodecamer in the absence of nucleotides. The p97^R155H^ hexamer does not stably bind to p47 in the absence of nucleotides. Once p97^R155H^ binds nucleotides, it stably interacts with p47. The p47 binding blocks the p97^R155H^ ATPase activity. Although the NTDs are all above the D1 ring plane in the presence of ADP, they adopt slightly different tilting angles and heights. However, in the presence of ATP, the NTDs are all in the same up configuration. The p47 interacts with the up-NTD, but not the down-NTD. Thicker lines indicate favor direction for reaction.

## Data Availability

Cryo-EM density maps have been deposited in the Electron Microscopy Data Bank (EMDB) under accession numbers EMD-24302 (p97^R155H^|_ATP_*_γ_*_S_-p47), EMD-24305 (p97^R155H^-p47), EMD-23191 (p97^R155H^ dodecamer), EMD-24304 (p97^R155H^|_ADP_-p47), and EMD-23192 (p97^R155H^ dodecamer II). Model coordinates were deposited in the Protein Data Bank (PDB) under accession numbers 7R7S (p97^R155H^|_ATP_*_γ_*_S_-p47), 7R7U (p97^R155H^-p47), 7L5W (p97^R155H^ dodecamer), and 7R7T (p97^R155H^|_ADP_-p47). All data are available from the corresponding authors upon request.

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
