# Peer review of "Structural and Functional Analysis of Disease-Linked p97 ATPase Mutant Complexes"

_ijms, 2021, doi:10.3390/ijms22158079_

Round 1

Reviewer 1 Report

The manuscript of Nandi et al. addresses the longstanding question how the disease-related mutations in VCP/p97 mechanistically affect the ATPase activity. For this, the authors show that purified p97 with R155H mutation can adopt a double-hexameric conformation and solved its structure by cryo-EM. Moreover, they build on their previous research to further analyze the effect of the adaptor protein p47 on the p97 ATPase rate. Using combinations of the R155H mutation and arginine finger mutants of p97, they conclude that the arginine fingers in p97 mediate both the inhibitory effect of p47 and the gain-of-function effect of the R155H mutation.

The manuscript includes some interesting results that contribute to the discussion of the effect of the p97 disease mutations on the p97 ATPase activity and how this is regulated by cofactors. This includes an interesting structure that demonstrates changes in the catalytic center caused by disease-mutation and/or p47 binding that is likely relevant for understanding the regulation of p97. However, the associated conclusions seem to lack experimental support. Moreover, the p97 dodecamer structure that they also present lacks novelty.

1) The overall structure of the dodecamer (Figure 2A) and in particular the interface between the two p97 hexamers (Figure 3) seems to be highly similar - if not identical - to the interface in the previously published structure of the p97-WT double hexamer. However, the recent publication was not cited (Hoq et al., ACS Nano 2021).

Moreover, since both p97-WT and p97-R155H exhibit ~40% of particles in double-hexameric conformation, the conclusion that the double-hexameric form depends on the R155H mutation does not seem to be supported by the data. In addition, the interpretation that the double-hexamer is a storage form in the abstract of the paper seems far-fetched, given that this conformation has so far only been observed in vitro, while the hexamer is the only form found in cellular analyses.

2) In Figure 4C, the authors present ATPase rates of p97 mutants and conclude that the arginine fingers are important for the gain-of-function effect of the R155H mutation, which is not supported by the data. First, interpretations and comparisons of the catalytically almost inactive R635A mutants does not seem meaningful. Second, the relevance of the observed differences in the presence of a substrate are questionable, since ATPase of p97 disease mutants rates are much less affected in the presence of a ubiquitin adaptor and/or substrate (Blythe et al., Structure 2019).

3) In Figure 4D, the authors functionally connect the proposed effects of R155H and p47 on the arginine finger to a biphasic ATPase regulation mechanism by p47, which they discovered previously. There is comments on this approach. First, it is difficult to understand how the authors can conclude that phase 2 activation is associated with D2 arginine fingers based on the data presented in this manuscript, when both R359A and R635A mutants show only phase 2 activation upon addition of p47. Second, when the R155H mutation leads to a decreased activity in phase 2 and arginine finger mutations lead to a (comparably strong) increased activity in phase 2, it is not surprising that the effect is nullified when the mutations are combined. It is difficult to follow the conclusion of a functional connection of p47 and the arginine fingers.

I have some additional minor points that should be addressed:

4) The Superdex 200 column used for gel filtration is not suitable to separate the p97 hexamer from the dodecamer. It seems that in particular the M1 fraction shown in Figure 1B may run in the void volume and may represent aggregated p97-R155H instead of dodecameric p97-R155H. It is therefore difficult to conclude from the absence of p47 in that fraction that p47 cannot bind to the dodecamer. To make that conclusion, the run should be repeated on a different column.

5) The Kd values presented in Figure 1D are not consistent with the data presented in Table S1. Moreover, the Kd values of p97-WT (97±32 µM) and p97-R155H (140±45 µM) are the same within the range of error. The authors can therefore not conclude that “p47 has a lower binding affinity to p97R155H mutant over the wild type p97 ATPase” (line 158). Also, I assume that the values displayed in table S1 are in mM (not µM, as stated).

6) In Figure 4A, the authors show two UBX domains of p47 on a p97 hexamer. Can the authors be sure that this is not an artifact due to the overlap of two conformations in which one UBX domain is bound to either the A protomer or the B protomer of the p97 hexamer? Are there enough particles that clearly show two UBX domains on one p97 hexamer?

7) In Figure S10, there is a clear p47 peak in fractions 18 and 19 for the R155H mutant, but there is no p47 peak for the R155HR359A double mutant. Thus, the authors should be careful to state that “p97R155H-R359A retained the ability to bind p47” (line 407). The p97-R155H-R359A mutant rather seems to run in the void volume and could be aggregated. Again, the Superdex 200 column is not suitable to separate complexes of this size.

8) The double mutants presented in Figure S11, in particular the p97R155H-R635A mutant, have a weaker affinity for p47 than p97R155H. How can the authors conclude that "the loss of activities of the p97R155H was not solely due to an inability of p97R155H to form a complex with p47" (lines 411-412)?

9) The authors state that “this observation of the high structural variabilities is consistent with the deep coordinate neural network analysis (Figure S9)” (line 197). The outcome of the deep coordinate neural network analysis is described neither in the text nor in the figure legend. Can the authors explain the analysis and the outcome? How can they conclude that the N-terminal domain is more flexible in the R155H mutant compare to the wild-type p97 (line 196), when there is no data for the wild-type p97 in the analysis?

10) In Figure 4D, it seems that 0.5 nM of p47 can inhibit 4 nM of p97 hexamer by about 40 %. Can the authors explain how one molecule of p47 can inhibit more than one p97 hexamer?

11) The authors state that “The structural superpositions showed that the change of the ATPase function may occur in a slight conformational change of the arginine finger via domain-domain communications (Figure 4A)” (lines 399-400). Where can these domain-domain communications that lead to the change of ATPase function be seen in the structure?

12) There is a typo in line 469: closed -> close

Reviewer 2 Report

Nandi et al. report the functional and structural analyses of the p97R155H, which is the most commonly found in the IBMPFD (inclusion body myopathy with Paget’s disease of bone and frontotemporal dementia) phenotype. The overall research flow and results are scientifically robust and are expected to potentially provide valuable insight into diseases related to p97 ATPase function studies. I support publication of the manuscript after the author has corrected the parts that should be clarified in the manuscript.

  1. The authors observed various oligomeric states in this experiments. Does this not change the proportion of oligomeric states according to each experimental batch? What is the difference in oligomeric state ratio compared to WT?

  2. I went to the PDB site to check the quality of the structure, but the deposited structure is in AUTH (processed, waiting for author review and approval), not HPUB (processing complete, entry on hold until publication). Authors should submit a validation report at revision to confirm the quality of the structure.

Minor

In the abbreviation, some words are underlined and some are not. The author should unify the expression into one.

e.g.

- SEP (shp1, eyc, and p47)
- transitional endoplasmic reticulum (TER)
- IBMPFD (inclusion body myopathy with Paget’s disease of bone and frontotemporal dementia)
- familial amyotrophic lateral sclerosis (ALS)

Round 2

Reviewer 1 Report

The authors significantly improved the paper by addressing all critical concerns.

In Figure S10, the image for the p97 blot seems to be missing, possibly due to software issues during figure composition. The authors might want to add this image for the final version of this article.

I have no further concerns and am in favor of publication.

Reviewer 2 Report

The revisions made in the manuscript have significantly improved the quality and readability of the manuscript. The responses to my specific comments and concerns are also satisfactory. I recommend acceptance of the revised manuscript for publication.